## Research Article

# Distinct groups of autoantigens as drivers of ocular adnexal MALT lymphoma pathogenesis

Richard J Bende[1,2,3,*], Naomi Donner[1,*], Thera AM Wormhoudt[1,2,3], Anna Beentjes[1], Angelique Scantlebery[1], Marloes Grobben[4,5], Khadija Tejjani[4,5], Felicity Chandler[6], Reina S Sikkema[6], Anton W Langerak[7], Jeroen EJ Guikema[1,2,3], Carel JM van Noesel[1,2,3]

Chronic B-cell receptor signals incited by cognate antigens are believed to play a crucial role in the pathogenesis of mucosa-associated lymphoid tissue lymphomas. We have explored the immunoglobulin variable regions (IGHV) expressed by 124 ocular adnexal MALT lymphomas (OAML) and tested the in vitro reactivity of recombinant IgM derived from 23 OAMLs. Six of 124 OAMLs (5%) were found to express a high-affinity stereotyped rheumatoid factor. OAMLs have a biased IGHV4-34 usage, which confers intrinsic super auto-antigen reactivity with poly-N-acetyllactosamine (NAL) epitopes, present on cell surface glycoproteins of erythrocytes and B cells. Twenty-one OAMLs (17%) expressed IGHV4-34-encoded B-cell receptors. Five of the 23 recombinant OAML IgMs expressed IGHV4-34, four of which bound to the linear NAL i epitope expressed on B cells but not to the branched NAL I epitope on erythrocytes. One non-IGHV4-34-encoded OAML IgM was also reactive with B cells. Interestingly, three of the 23 OAML IgMs (13%) specifically reacted with proteins of U1-/U-snRNP complexes, which have been implicated as cognate-antigens in various autoimmune diseases such as systemic lupus erythematosus and mixed connective tissue disease. The findings indicate that local autoimmune reactions are instrumental in the pathogenesis of a substantial fraction of OAMLs.

## Introduction

Extranodal marginal zone lymphomas of mucosa-associated lymphoid tissue (MALT) develop at locations of chronic inflammation because of infections or autoimmune diseases such as *Helicobacter pylori* (*Hp*) gastritis, Sjögren's sialo-adenitis, and Hashimoto's thyroiditis (Bende et al, 2009; Schreuder et al, 2017; Du, 2020; Rossi et al, 2022). The sustained inflammatory environment supports formation of durable organized lymphoid tissue, including secondary lymphoid follicles (Bende et al, 2009). In these structures, antigen-responsive B cells with help of T cells may repeatedly engage in germinal center reactions resulting in extensive clonal expansion and accumulation of (epi)genetic alterations with the inherent risk of malignant transformation (Bende et al, 2007; Pelissier et al, 2023).

With respect to the immune repertoire of MALT lymphomas, rheumatoid factor (RF) activity (auto-reactivity with IgG) was found as a frequent, and thus far as the only, established antigenic specificity of MALT lymphomas (Martin et al, 2000; Bende et al, 2005, 2020). These MALT lymphoma RFs, most often encoded by stereotyped RF immunoglobulin (IG) gene rearrangements, are expressed by 70–80% of Sjögren's syndrome-associated salivary gland MALT lymphomas and frequently by gastric MALT lymphomas and hepatitis C virus-related B-cell lymphomas (De Re et al, 2000; Bende et al, 2005, 2020). In two other studies on seven gastric- and five OAML, of which the expressed BCRs were produced as IgG, only polyreactivity with various self and non–self-antigens was demonstrated (Craig et al, 2010; Zhu et al, 2015).

Apart from providing B-cell antigen receptor (BCR) ligands, the role of the microenvironment is multipart, and, for example it also involves ligation of pattern recognition receptors and CD40 by CD40L-expressing T cells. The significance of environmental factors in supporting tumor cell growth is underscored by the fact that most gastric MALT lymphomas are cured by *Hp* eradication (Wotherspoon et al, 1993; Fischbach et al, 2007; Ruskone-Fourmestraux et al, 2011; Nakamura et al, 2012; Zucca et al, 2020). In MALT lymphomas at other primary sites such as ocular adnexa, lung, and more rare locations, the etiology and role of the chronic inflammation remain unclear (Das et al, 2019; Derakhshandeh et al, 2021). In some studies, an association between ocular adnexal MALT

[1]Department of Pathology, Amsterdam UMC, Location University of Amsterdam, Amsterdam, Netherlands [2]Lymphoma and Myeloma Center (LYMMCARE), Amsterdam, Netherlands [3]Cancer Center Amsterdam (CCA), Amsterdam, Netherlands [4]Department of Medical Microbiology and Infection Prevention, Laboratory of Experimental Virology, Amsterdam UMC, Location University of Amsterdam, Amsterdam, Netherlands [5]Amsterdam Institute for Infection and Immunity, Infectious Diseases, Amsterdam, Netherlands [6]Department of Viroscience, Erasmus MC, Rotterdam, Netherlands [7]Department of Immunology, Laboratory Medical Immunology, Erasmus MC, Rotterdam, Netherlands

Correspondence: c.j.vannoesel@amsterdamumc.nl; r.j.bende@amsterdamumc.nl
*Richard J Bende and Naomi Donner contributed equally to this work

lymphomas (OAML) and chronic *Chlamydia psittaci* infection was proposed, an association that could not be substantiated by other studies (Bende et al, 2009; Schreuder et al, 2017; Johansson et al, 2022).

MALT lymphomas of the various primary locations have dissimilar oncogenetic and microenvironmental drivers as reflected by specific genetic hallmarks and by expression of typical IG variable heavy chain (IGHV) genes suggesting other BCR specificities. The most well-known MALT lymphoma-specific translocation t(11; 18)(q21; q21), encoding a BIRC3-MALT1 gene fusion product, is frequent in lung (40%), intestine (35%) and gastric (20%) MALT lymphomas but very rare in salivary gland, ocular adnexal and thyroid MALT lymphomas (Bende et al, 2009). Inactivation of *TNFAIP3* by mutation or deletion occurs typically in OAMLs (40%) (Moody et al, 2018), whereas this is rare in other MALT lymphomas (Moody et al, 2018; Vela et al, 2022). Alternatively, *TBL1XR1* and *GPR34* mutations are frequent in salivary gland MALT lymphomas (24% and 19%) (Moody et al, 2018) and uncommon in other MALT lymphomas (Moody et al, 2018; Johansson et al, 2020; Magistri et al, 2021; Vela et al, 2022).

We and others have shown that 20% of OAMLs express functional IGHV4-34 rearrangements (Zhu et al, 2011; Dagklis et al, 2012; van Maldegem et al, 2012). Interestingly, IGHV4-34-expressing OAMLs were reported to be significantly associated with *TNFAIP3* inactivation (Moody et al, 2017). It has been established that IGHV4-34-encoded BCRs possess intrinsic super auto-antigen reactivity by variably binding with poly-N-acetyllactosamine (NAL) epitopes, which are present on cell surface glycoproteins and lipids of erythrocytes (adult I blood group determinant, branched NAL) and B cells (i determinant, linear NAL) (Bhat et al, 2000). This auto-reactivity requires two motifs within frame work region 1 (FR1), that is, $Q^6W^7$ and $A^{24}V^{25}Y^{26}$ (Li et al, 1996; Potter et al, 2002). Here, we present data on the repertoire and the antigenic specificity of the surface immunoglobulins expressed by OAMLs, uncovering subsets of OAMLs specific for distinct groups of auto-antigens.

# Results

### IG variable heavy and light chain expression of OAML

We have studied 124 immunoglobulin variable heavy (IGHV) gene sequences of a total of 121 OAML patients, that is, (i) 30 IGHV sequences and 16 corresponding IG light chain variable (IGKV/IGLV) sequences of 29 patients diagnosed at our department (Tables S1 and S2) and (ii) 94 IGHV and 24 corresponding IGKV/IGLV sequences of 92 patients, published in literature (Bahler et al, 2009; Zhu et al, 2011; Dagklis et al, 2012; Zhu et al, 2013). The IGHV4-34 gene was expressed by 21 of 124 (17%) OAMLs. In six of 11 (55%) IGHV4-34 OAMLs, of which the IGKV had also been identified, the IGHV4-34 was combined with a typical IGKV3-20-encoded IG light chain (Table S2) (Zhu et al, 2011, 2013; van Maldegem et al, 2012).

### VH-CDR3 amino acid sequence homology analysis of OAML

The OAML-derived VH-CDR3 amino acid (aa) sequences were analyzed for homology with VH-CDR3 aa sequences present in

GenBank (release 244), using the NCBI Protein-BLAST algorithm. VH-CDR3s were defined as being homologous when they shared at least 60% amino acid sequence homology, with an allowed maximal gap of three amino acids (Tables S3 and S4) (Bende et al, 2016, 2020, 2022).

Forty nine of 124 OAML VH-CDR3s (40%), showed homology with VH-CDR3s retrieved from B cells of inflamed tissues, for example, from synovium of rheumatoid arthritis patients, 60% with VH-CDR3s of anti-viral antibodies, 30% with VH-CDR3s expressed by B-cell lymphomas, of which 20% were classified as chronic lymphocytic leukemia (CLL) (Fig 1A). Interestingly, 17 OAML VH-CDR3s were homologous with CLLs belonging to so-called stereotyped BCR subsets as defined by the European Research Initiative on CLL (ERIC) consortium (Table S5)(Agathangelidis et al, 2021).

Four OAMLs showed limited intra-OAML VH-CDR3 homology (Table S6). A remarkable finding was the identification of two IGHV4-34-expressing cases (OAML 38706 and OM12) with near identical VH-CDR3 and sharing eight somatic replacement mutations in the region of aa 48 of VH-FR2 to VH-CDR3 of IGHV (Fig 1C). Otherwise, we did not identify (a) group(s) of OAML with significant intra OAML VH-CDR3 homology.

Six of 124 OAML (5%) expressed a stereotyped RF BCR, five of which were belonging to one of three stereotyped RF groups. Previously, we repeatedly confirmed that recombinantly produced IgMs of these three groups indeed displayed strong mono-reactive binding to IgG (Fig 1B) (Bende et al, 2005, 2016, 2020; Hoogeboom et al, 2010, 2013b; Janssen et al, 2021). One case (OAMZL44) was unique as it expressed an IGHV3-74-encoded BCR, which was highly homologous to an RF-producing memory B-cell clone (D1 P3) that we had isolated by antigen-specific sorting with labeled IgG of a peripheral blood mononuclear cell sample of a donor who had been immunized with Rhesus(D)-mismatched red blood cells (Fig 1B). Donors immunized with mismatched red blood cells are known to frequently harbor B cells expressing stereotyped RF BCRs (Borretzen et al, 1995, 1997).

### Expression of recombinant OAML IgM

We produced recombinant IgM of 23 OAMLs, 14 were from our own cohort and 9 derived of OAMLs reported by Zhu et al (2011, 2013). Five of the 23 OAML IgMs were encoded by IGHV4-34 rearrangements, four (80%) of which co-expressed an IGKV3-20-encoded light chain (Table 1). Of the IGHV3-74-expressing OAMZL44 only the IGHV sequence was reported (Fig 1B). Therefore, we choose to combine the OAMZL44 IGHV3-74 sequence with the IGKV4-1/JK4-encoded light chain of the homologous RF-expressing D1 P3 memory B-cell clone.

### Binding of OAML IgM to B-cell lines and erythrocytes

Because OAML BCRs are biased towards expression of IGHV4-34, known to confer intrinsic super auto-antigen reactivity with poly-N-acetyllactosamine (NAL) epitopes present on B cells and erythrocytes (Li et al, 1996; Bhat et al, 2000; Potter et al, 2002), we tested the 23 OAML IgMs for binding to (i) an IgG-expressing EBV transformed memory B-cell line LOS2 (Bende et al, 1992), (ii) an IgG-expressing diffuse large B-cell lymphoma (DLBCL) cell line

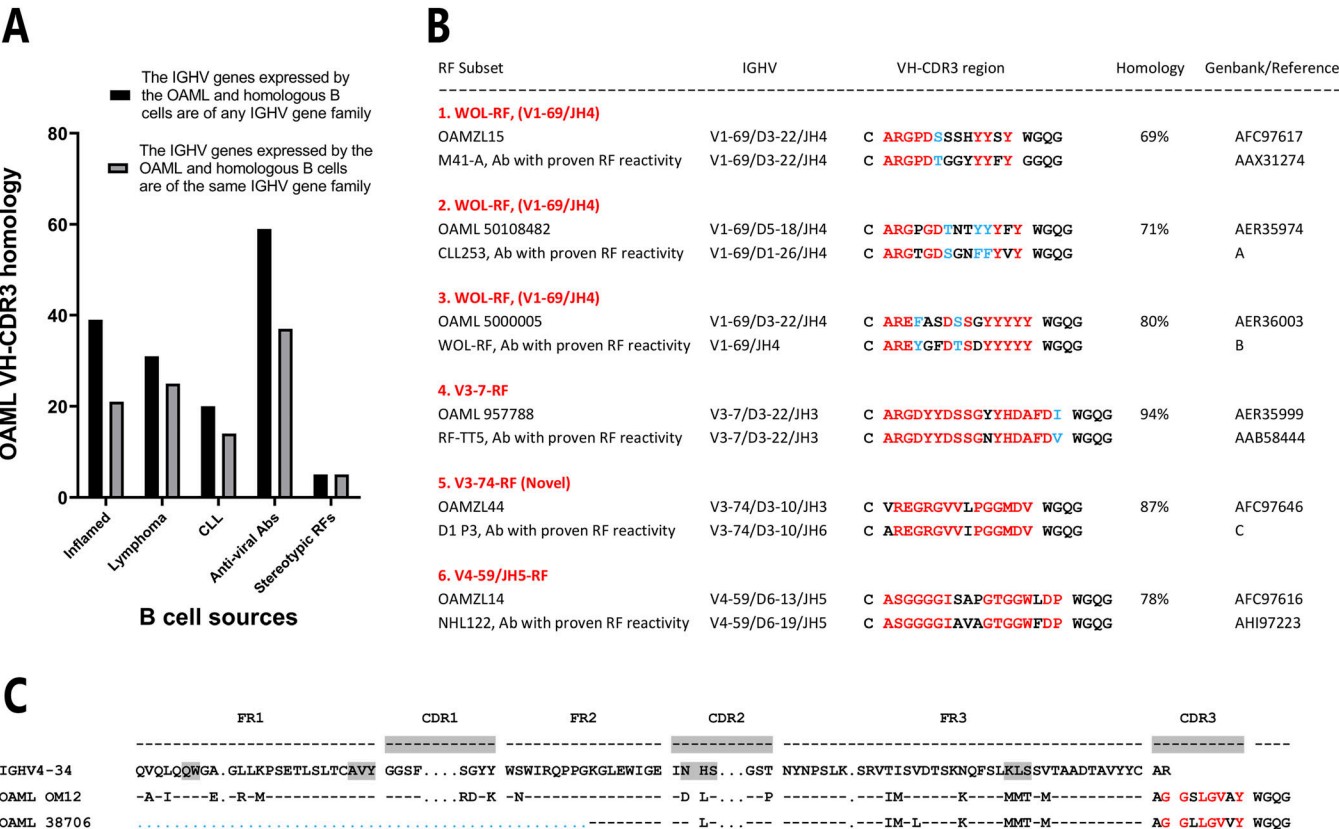

**Figure 1. VH-CDR3 homology analysis of 124 OAMLs.**
**(A)** Percentage of OAMLs having VH-CDR3 homology with VH-CDR3 sequences from GenBank, derived from (i) various sites of inflammation, (ii) B-cell lymphomas, (iii) CLLs, (iv) anti-viral antibodies, and (v) stereotypic RFs (detailed information in Tables S2 and S3). **(B)** IGHV-CDR3 homology between six OAMLs with stereotyped RFs with proven in vitro RF reactivity. Identical and similar amino acids are highlighted in red and blue, respectively. Reference A: PMID 33778415 (Janssen et al, 2021), Reference B: PMID: 2439644 (Newkirk et al, 1987), Reference C: Bende, unpublished. **(C)** IGHV amino acid sequences of two homologous IGHV4-34-expressing OAMLs.

LY3, and (iii) Rhesus(D) positive- and negative-erythrocytes. As additional controls we included three IGHV4-34-encoded IgMs derived from CLL.

As measured by flowcytometry, five of the OAML IgMs bound variably to the IgG-expressing B-cell lines. Binding to erythrocyte appeared much weaker or was not detectable at all. OM12 and OM30 IgM both bound strongly to the LOS2 and LY3 cells, OM24 showed stronger binding to LY3, whereas OM40 and OAMZL10 IgM displayed stronger binding to the LOS2 cells. The OM12 IgM also bound weakly to both Rhesus(D) positive and negative erythrocytes (Fig 2A and B).

Of the five IgMs that bind to B-cell lines, four (OM12, OM24, OM40, and OAMZL10) were encoded by IGHV4-34 rearrangements, whereas the OM30 IgM was encoded by an IGHV4-61 rearrangement. The IGHV4-34-expressing OAMZL4 did not show binding, which is explained by an A24D mutation in the VH-FR1 $Q^6W^7/A^{24}V^{25}Y^{26}$ motif, which is critical for NAL binding (Fig S1).

We produced two variant IgMs of OM12, one in which the somatic mutations of IGHV were reverted to their respective germline IGHV sequences (OM12VH-GL) and one, in which $W^7$ was mutated in $S^7$ in the VH-FR1 NAL-binding motif (OM12-W7S). Interestingly, the OM12VH-GL IgM showed stronger binding to B cells and erythrocytes, whereas the OM12-W7S, as expected, had lost its binding to B cells and erythrocytes (Fig 2C). The fact that the somatic IGHV

mutations in OM12 IgM resulted in lower binding capacities to B cells and erythrocytes as compared with the OM12VH-GL IgM indicates that during affinity maturation of OM12 IgM, selection for binding to a particular antigen simultaneously resulted in lower binding strengths to NAL epitopes.

The IgMs of the control IGHV4-34-expressing CLLs and of a poly-reactive CLL bound very well to the LOS2 and LY3 cell lines. The CLL195 IgM also bound strongly with and even induced some ag-glutination of in particular Rhesus(D) negative erythrocytes (Fig 2D).

### Antigenic specificity of the OAML-derived IgM

To further delineate the antigen specificities, the 23 OAML IgMs were tested in ELISA for binding to established self/auto-antigens, fungal and bacterial antigens and for binding to viral antigens in a Luminex assay and in an antigen array (antigens listed in Table S7). In addition, the OAML IgMs were tested using immunohisto-chemistry for binding to a micro-array of various paraffin-embedded normal human tissues (TMA).

Thirteen OAML IgMs did not display significant binding in any of these assays. Three OAML IgMs (OM23, OM30, and OAMZL16) showed weak poly-reactivity with antigens such as insulin, small nuclear ribonucleoprotein (snRNP)-B, C, 68/70, and SSA/Ro52, which,

**Table 1.** IGHV and IGKV/IGLV rearrangments of 23 recombinantly produced OAML IgM antibodies.

| Patient | IGHV rearrangement | No. of mut. | VH-CDR3 (aa) | IGKV/IGLV rearrangement | No. of mut. | VK/VL-CDR3 (aa) |
|---|---|---|---|---|---|---|
| OM3B | V4-31/D3-10/JH4 | 18 | C ARLSGSGNYHDYGRFDS WGQG | VK2-28/JK4 | 5 | C MQALQTPLT FGGG |
| OM8 | V3-23/D2-2/JH6 | 28 | C AKGQLREMKYYYYGMDV WGQG | VK2D-29/JK2 | 8 | C MQSIQLPPMT FGQG |
| OM9B | V3-9/D2-15/JH4 | 4 | C AKDSGDNRCYPSSSAWCGVDY WGQG | VK4-1/JK4 | 2 | C QQYYSTPQT FGQG |
| OM12 | **V4-34/D2-8/JH4** | 23 | C AGGSLGVAY WGQG | **VK3-20/JK2** | 17 | C QHYGSSPYT FGQG |
| OM23 | V4-59/D3-9/JH4 | 17 | C ARQRGGGGYDIFTGSSHFFVH WGQG | VK1-8/JK1 | 23 | C QQHYDFPAT FGLG |
| OM24 | **V4-34/D2-15/JH3** | 11 | C ASPGYCSGGSCYPNGFDI WGQG | VK3-11/JK2 | 6 | C QQRSNWPYT FGQG |
| OM30 | V4-61/D2-8/JH3 | 8 | C AREVFDAFDI WGQG | VK3-20/JK2 | 8 | C QHYRRSPYT FGQG |
| OM31 | V4-30.4/D4-23/JH4 | 8 | C ARELRGSSVEY WGQG | VK3-15/JK1 | 3 | C QQYHNWPPWT FGQG |
| OM38 | V3-23/D6-19/JH5 | 16 | C AKGGSGWPTPSFF WGQG | VK1-39/JK4 | 21 | C QQTYSPPLT FGGG |
| OM40 | **V4-34/D6-6/JH5** | 12 | C ARGPGYDNSSPA WGQG | **VK3-20/JK4** | 2 | C QQYGSSPLT FGGG |
| OM46 | V3-30/D1-20/JH4 | 11 | C ATGPLEIITGTTLNY WGQG | VK3-20/JK2 | 9 | C QKYDSSPYT FGQG |
| OM56 | V1-69/D2-21/JH3 | 30 | C ARHISLHFNGGPFDI WGLG | VL2-14/JL1 | 30 | C QSYDERLGGWV FGGG |
| OM66 | V4-4/D4-23/JH6 | 30 | C AGTYSDYGSYFAYYMDV WGQG | VK3-20/JK4 | 10 | C QQYGSSVLT FGGG |
| OM71 | V3-11/D3-16/JH4 | 22 | C ARALGGRIAPFDF WGQG | VK1-5/JK2 | 22 | C QQYSDFPYN FGQG |
| OAMZL1 | V3-74/D1-26/JH4 | 7 | C ARVGVGAYDY WGQG | VK4-1/JK1 | 1 | C QQYYSTWT FGQG |
| OAMZL3 | V3-11/D6-19/JH4 | 13 | C ARQGSEYSSGWYMATDY WGQG | VK3-15/JH2 | 4 | C QQYNNWPRYT FGQG |
| OAMZL4 | **V4-34/D6-19/JH4** | 64 | C ARVNQGLLDS WGHG | **VK3-20/JK2** | 18 | C QQYRSSPVT FGQG |
| OAMZL9 | V3-23/D3-3/JH5 | 26 | C ANWSSPYPTWFDP WGQG | VK1-8/JK5 | 13 | C QQYYNYPIT FGQG |
| OAMZL10 | **V4-34/D2-2/JH6** | 29 | C ARATLVPASIVYRTHYYSGIDV WGQG | **VK3-20/JK1** | 10 | C QQYGTSPRT FGQG |
| OAMZL13 | V3-30/D1-20/JH6 | 15 | C AKARSVWSPGHYDMDV WGQG | VK1-33/JK5 | 7 | C QQHDNLPIT FGQG |
| OAMZL16 | V3-66/D3-22/JH4 | 9 | C AGAWVYDSSVFDY WGQG | VK1-5/JK4 | 1 | C QQYNTYPLT FGGG |
| OAMZL21 | V1-24/D5-24/JH4 | 18 | C VAAVGDGHNYFDS WGQG | VK4-1/JK4 | 11 | C QQYYTTPPT FGGG |
| OAMZL44 | V3-74/D3-10/JH3 | 26 | C VREGRGVVLPGGMDV WGQG | VK4-1/JK2 [a] | 2 | C QQYYSTPYT FGGG |

[a]The V3-74-encoded IGHV of OAMZL44 was combined with the VK4-1-encoded IGKV of the highly homologous D1 P3 B-cell clone (see Fig 1B).

however, was substantially lower than that of two selected poly-reactive IgMs derived from IGHV-unmutated CLLs (U-CLL) (Fig 3). Three of the four IGHV4-34-expressing B-cell binding OAML IgMs (OM12, OM40, and OAMZL10) also demonstrated binding to B cells in spleen tissue. All other OAML IgMs did not stain any cell types in the TMA (data not shown). The IgMs of OM8, OAMZL3, and OAMZL44 demonstrated strong mono-specific RF activity (Fig 4A). The BCRs of OM8 and OAMZL3 are not encoded by stereotyped RF IGHV rearrangements, whereas OAMZL44 is encoded by a here newly defined stereotyped RF IGHV3-74/D3-10/JH3 rearrangement, encoding a characteristic VH-CDR3 amino acid sequence, which is combined with an IGKV4-1-encoded light chain (Fig 1B).

Interestingly, OM12, OM56 and OAMZL21 IgM specifically bound to proteins of the U1-/U-snRNP complexes, that is, U-snRNP-B/B' and U1-snRNP-C. In addition, OAMZL21 IgM also bound to U1-snRNP-68/70 (Fig 4B–D). We produced variant IgMs of OM12, OM56, and OAMZL21, in which the somatic mutations of their IGHVs were reverted into their respective germline (GL) IGHV sequences. These GL-reverted IgMs demonstrated a complete loss of binding affinity for the proteins of the U1-/U-snRNP complexes, showing that the somatic IGHV mutations had been affinity-selected (Fig 4B–D). The

fact that OM12, OM56, and OAMZL21 all show crossreactivity between U-snRNP-B/B' and U1-snRNP-C, and OAMZL21 also with U1-snRNP-68/70 can be explained by the homology between these proteins as shown in Fig S2.

Of note, of the produced OAML IgMs, nine showed VH-CDR3 homology with antibodies specific for viral antigens, which were included in our antigen binding screenings, that is, respiratory syncytial virus F protein, influenza virus HA protein and severe acute respiratory syndrome corona virus S protein (Tables S3 and S4). Unfortunately, all these IgMs were not reactive in the various binding assays for these viral antigens (Table S8).

### IgM derived of two CLL stereotyped IGHV groups that are IGHV homologous with two OAMLs show binding with U1-/U-snRNP complex proteins

We identified 17 OAML VH-CDR3s, each of which were homologous with an ERIC-defined CLL stereotyped BCR subset (Table S5). Unfortunately, because of most of these 17 OAMLs only a part of the IGHV sequences and no IGKV/IGLV sequences are available in literature, we were unable to produce their antibodies

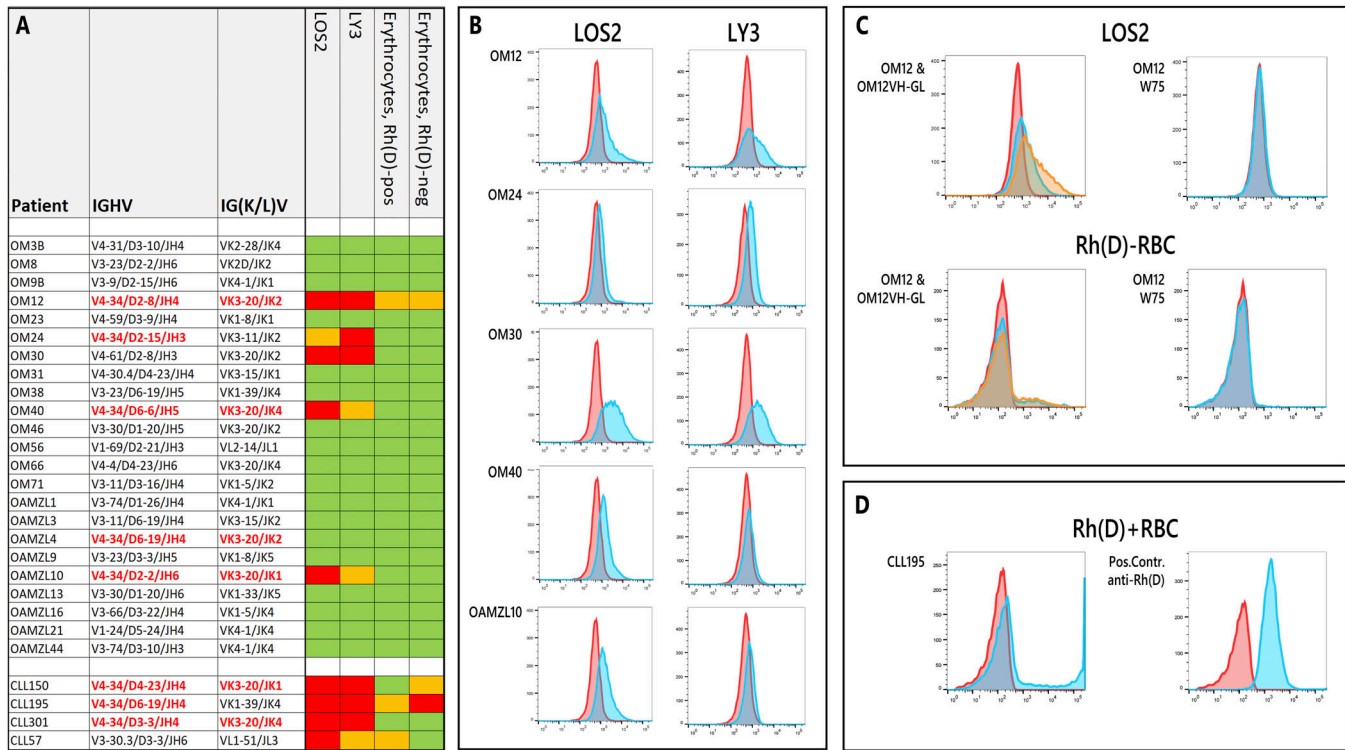

**Figure 2. Binding profiles of OAML IgMs to B cells and erythrocytes.**
**(A)** Binding capacity of the 23 recombinant OAML IgMs with the IgG-expressing B-cell lines LOS2 and LY3 as well as to Rhesus(D)-positive and -negative erythrocytes. Red, orange, and green indicate strong, moderate, and no binding, respectively. The cut-offs for moderate and strong binding were, respectively, >1.2 times and >1.5 times background geometric mean fluorescence intensity. **(B)** Flow cytometry graphs depicting OAML IgM reactivity with the IgG-expressing B-cell lines LOS2 and LY3 in blue. Red curves cells depict non-binding human IgM control antibody. **(C)** Left graphs, binding of OM12 IgM in blue and OM12VH-GL IgM in orange to LOS2 cells and Rhesus(D)-negative red blood cells. Right graphs depict absent binding of OM12 W7S IgM to LOS2 cells and Rhesus(D)-negative red blood cells. **(D)** Binding of the CLL195-derived IgM and a control human anti-Rhesus(D) monoclonal IgM to Rhesus(D)-positive red blood cells.

recombinantly. However, primary CLL tumor cells of some OAML IGHV-homologous CLLs of the ERIC consortium were obtained, that is, one CLL of stereotyped subset #31, three of #148B, two of #202 and one of #273 (Fig 5A). These primary CLL cells were stimulated in culture to secrete CLL-derived IgM, as described previously (Hoogeboom et al, 2015; Janssen et al, 2021). Of note, OAMZL12 VH-CDR3 is >65% homologous with CLL subset #148B members, for example, DE-02-0294-H1, ID84, and D33 (Table S3). However, the OAMZL12 VH-CDR3 shares <60% VH-CDR3 homology with the genuine ERIC #148B subset members CLL406, CLL409, and CLL410 that we could obtain for our experiments (Fig 5A).

The two IgMs of subset #202 and the IgM of subset #273 did not show any binding with the antigens in ELISA. Interestingly, IgM of the subset #31 CLL, (CLL403) homologous with OAMZL26, showed reactivity with the proteins 68/70, A, B, and C of the U1-/U-snRNP complexes. The IgMs secreted by three subset #148B CLLs (CLL406, CLL409, CLL410), homologous with OAMZL12, showed binding to the A, B and C proteins of the U1-/U-snRNP complexes to a varying degree (Fig 5B).

## Discussion

In this study, we analyzed 124 VH-CDR3 amino acid sequences of OAMLs derived from 121 patients suffering from OAML and tested the in vitro reactivity of 23 recombinantly produced OAML IgMs. Three of the 23 OAML IgMs showed weak poly-reactivity in vitro, in accordance with a previous study on five OAMLs (Zhu et al, 2015). The level of poly-reactivity of the three OAML IgMs, however, was substantially lower than that of two selected poly-reactive IgMs retrieved from U-CLLs. Six of the 124 OAML IGHVs (5%) were stereotyped high affinity RFs (Bende et al, 2005, 2016, 2020, 2022; Hoogeboom et al, 2010, 2013b; Janssen et al, 2021). Three of the produced 23 (13%) OAML IgMs displayed strong mono-reactive RF activity in an IgG ELISA. The IgMs of OM8 and OAMZL3 were not encoded by stereotyped RF IGHV rearrangements, whereas OAMZL44 expressed the here newly defined stereotyped RF IGHV3-74/IGKV4-1 combination. RF mono-reactivity has previously also been demonstrated by us for an immunological subset of M-CLL (Hoogeboom et al, 2013b; Janssen et al, 2021; Bende et al, 2022). However, in contrast to our previous findings on subsets of M-CLL, we did not observe binding of OAML IgMs with any viral, bacterial, or fungal antigens (Hoogeboom et al, 2013a; Bende et al, 2022).

Twenty-one of the 124 OAMLs (17%) expressed IGHV4-34-encoded BCRs (Zhu et al, 2011; Dagklis et al, 2012; van Maldegem et al, 2012). A bias towards IGHV4-34 expression is not unique for OAMLs and is also reported for 30% of activated B-cell (ABC) DLBCL (Young et al, 2015), 30% of primary central nervous system lymphomas (Montesinos-Rongen et al, 2014), 60% of primary vitreoretinal lymphomas

| Patient | IGHV | IG(K/L)V | Conclusion Reactivity | Self/Auto Antigens | | | | | | | | | | | | IA | | |
|---|---|---|---|---|---|---|---|---|---|---|---|---|---|---|---|---|---|---|
| | | | | Actin | Insulin | Human IgG (RF-activity) | SSA/Ro52 | SSA/Ro60 | SSB/La48 | U1-snRNP-68/70 | U1-snRNP-A | U-snRNP-B/B' | U1-snRNP-C | U-snRNP-D | 7 other Auto Antigens | 12 fungal/Bacterial Antigens | 10 viral Antigens (Luminex) | 17 viral Antigens (Array) |
| OM3B | V4-31/D3-10/JH4 | VK2-28/JK4 | Non/Weak reactive | | | | | | | | | | | | | | | |
| OM8 | V3-23/D2-2/JH6 | VK2D/JK2 | **RF activity** | | | R | | | | | | | | | | | | |
| OM9B | V3-9/D2-15/JH6 | VK4-1/JK1 | Non/Weak reactive | | | | | | | | | | | | | | | |
| OM12 | **V4-34/D2-8/JH4** | **VK3-20/JK2** | **U-snRNP reactive** | | | | | | | | | R | R | | | | | |
| OM23 | V4-59/D3-9/JH4 | VK1-8/JK1 | **Weak Polyreactive** | | O | | | | | | | O | | | | | | |
| OM24 | **V4-34/D2-15/JH3** | VK3-11/JK2 | Non/Weak reactive | | | | | | | O | | | | | | | | |
| OM30 | V4-61/D2-8/JH3 | VK3-20/JK2 | **Weak Polyreactive** | | | | | | | O | | O | O | | | | | |
| OM31 | V4-30.4/D4-23/JH4 | VK3-15/JK1 | Non/Weak reactive | | | | | | | | | | | | | | | |
| OM38 | V3-23/D6-19/JH5 | VK1-39/JK4 | Non/Weak reactive | | O | | | | | | | | | | | | | |
| OM40 | **V4-34/D6-6/JH5** | **VK3-20/JK4** | Non/Weak reactive | | | | | | | | | | | | | | | |
| OM46 | V3-30/D1-20/JH5 | VK3-20/JK2 | Non/Weak reactive | | | | | | | | | | | | | | | |
| OM56 | V1-69/D2-21/JH3 | VL2-14/JL1 | **U-snRNP reactive** | | | | | | | O | | R | R | | | | | |
| OM66 | V4-4/D4-23/JH6 | VK3-20/JK4 | Non/Weak reactive | | | | | | | | | | | | | | | |
| OM71 | V3-11/D3-16/JH4 | VK1-5/JK2 | Non/Weak reactive | | | | | | | | | | | | | | | |
| OAMZL1 | V3-74/D1-26/JH4 | VK4-1/JK1 | Non/Weak reactive | | | | | | | | | | | | | | | |
| OAMZL3 | V3-11/D6-19/JH4 | VK3-15/JK2 | **RF activity** | | O | R | | | | | | | | | | | | |
| OAMZL4 | **V4-34/D6-19/JH4** | **VK3-20/JK2** | Non/Weak reactive | | | | | | | | | | | | | | | |
| OAMZL9 | V3-23/D3-3/JH5 | VK1-8/JK5 | Non/Weak reactive | | | | | | | | | | | | | | | |
| OAMZL10 | **V4-34/D2-2/JH6** | **VK3-20/JK1** | Non/Weak reactive | | | | | | | | | | | | | | | |
| OAMZL13 | V3-30/D1-20/JH6 | VK1-33/JK5 | Non/Weak reactive | | | | | | | | | | | | | | | |
| OAMZL16 | V3-66/D3-22/JH4 | VK1-5/JK4 | **Weak Polyreactive** | | | | O | | | O | | | | | | | | |
| OAMZL21 | V1-24/D5-24/JH4 | VK4-1/JK4 | **U-snRNP reactive** | | | | | | | R | | R | R | | | | | |
| OAMZL44 | V3-74/D3-10/JH3 | VK4-1/JK2 * | **RF activity** | | | R | | | | | | | | | | | | |
| M8 | V3-30/D3-9/JH5 | VK1-39/JK4 | Non/Weak reactive | | | | | | | | | | | | | | | |
| CLL57 (U) | V3-30.3/D3-3/JH6 | VL1-51/JL3 | **Strong Polyreactive** | R | R | R | R | R | R | R | R | R | R | R | R | R | R | R |
| CLL299 (U) | V1-2/D1-26/JH6 | VK4-1/JK2 | **Strong Polyreactive** | R | R | R | R | R | R | R | R | R | R | R | R | R | R | R |

**Figure 3. Binding profiles of OAML IgMs to self/auto antigens and antigens of infectious agents.**
Red and orange, respectively, indicates strong (>5 times background ABS 450 nm) and moderate (3–5 times background ABS 450 nm) antigen binding of the IgM in ELISA. Green indicates no antigen binding. IA = infectious agents. All 57 antigens tested are listed in Table S7.

(Belhouachi et al, 2020) and 80% of primary cold agglutinin disease patients (CAD) (Pascual et al, 1992; Malecka et al, 2016). It is well known that IGHV4-34-encoded antibodies may have intrinsic affinity for NAL epitopes and thus may bind to erythrocytes and B cells (Li et al, 1996; Bhat et al, 2000; Potter et al, 2002). However, as yet the actual reactivity of IGHV4-34-expressing lymphomas has been only demonstrated to a limited extent: (i) the BCR of the ABC DLBCL cell line HBL1 was shown to bind to its own cell surface (Young et al, 2015), (ii) three DLBCL-derived IgM were shown to variably bind to the human pre–B-cell line NALM6 and to cord blood (i determinant, linear NAL) and adult blood erythrocytes (I determinant, branched NAL) (Bhat et al, 2000) and (iii) of numerous CAD-derived IgMs preferential binding to the branched NAL I epitope on adult erythrocytes has been reported (Pascual et al, 1992; Malecka et al, 2016).

Among the 23 OAML IgMs produced, five were IGHV4-34-encoded of which OM12, OM24, OM40, and OAMZL10 bound to B cells. In addition, one non-IGHV4-34-encoded OAML IgM, the IGHV4-61/IGKV3-20-expressing OM30, also showed in vitro binding to B cells. The lack of binding to B cells of the IGHV4-34-expressing OAMZL4 is explained by an A24D mutation in the VH-FR1 $Q^6W^7/A^{24}V^{25}Y^{26}$ motif, known to be critical for NAL binding. In accordance, the W7S mutant of OM12 also demonstrated absent binding to B cells and erythrocytes. In OM12, OM40, and OAMZL10, the IGHV4-34 was combined with an IGKV3-20-encoded IGL chain. Of note, the combination of IGHV4-34 with IGKV3-20, or with the highly homologous IGKV3-15, is found in 90% of CAD patients and results in IgM antibodies with a relatively high affinity for the branched NAL I epitope on erythrocytes clinically leading to pathological red blood cell agglutination (Bhat et al, 2000; Malecka et al, 2016). In addition to the associated IGKV/IGLV gene, the binding strength of VH4-34-encoded antibodies to erythrocytes also depends on somatic IGHV mutations and VH-CDR3 composition (Li et al, 1996). Remarkably, OM12, OM40, and OAMZL10 each express IGHV4-34/IGKV3-20 IgM and all bound to B cells, but displayed no or a low affinity for the branched NAL I epitope on erythrocytes. Most likely, during the process of lymphomagenesis precursor cells were selected for binding to the linear NAL i epitope expressed on B cells, providing sustained BCR stimulation and obtaining a growth advantage within the tissue microenvironment.

Three of the 23 OAML IgMs (13%) specifically bound U1-/U-snRNP complexes; the IgM of OM12, OM56, and OAMZL21 bound to U-snRNP-B/B' and U1-snRNP-C, whereas OAMZL21 IgM also reacted with U1-snRNP-68/70. These reactivities were abolished by removal of the somatic IGHV mutations, implying that they were affinity-selected for

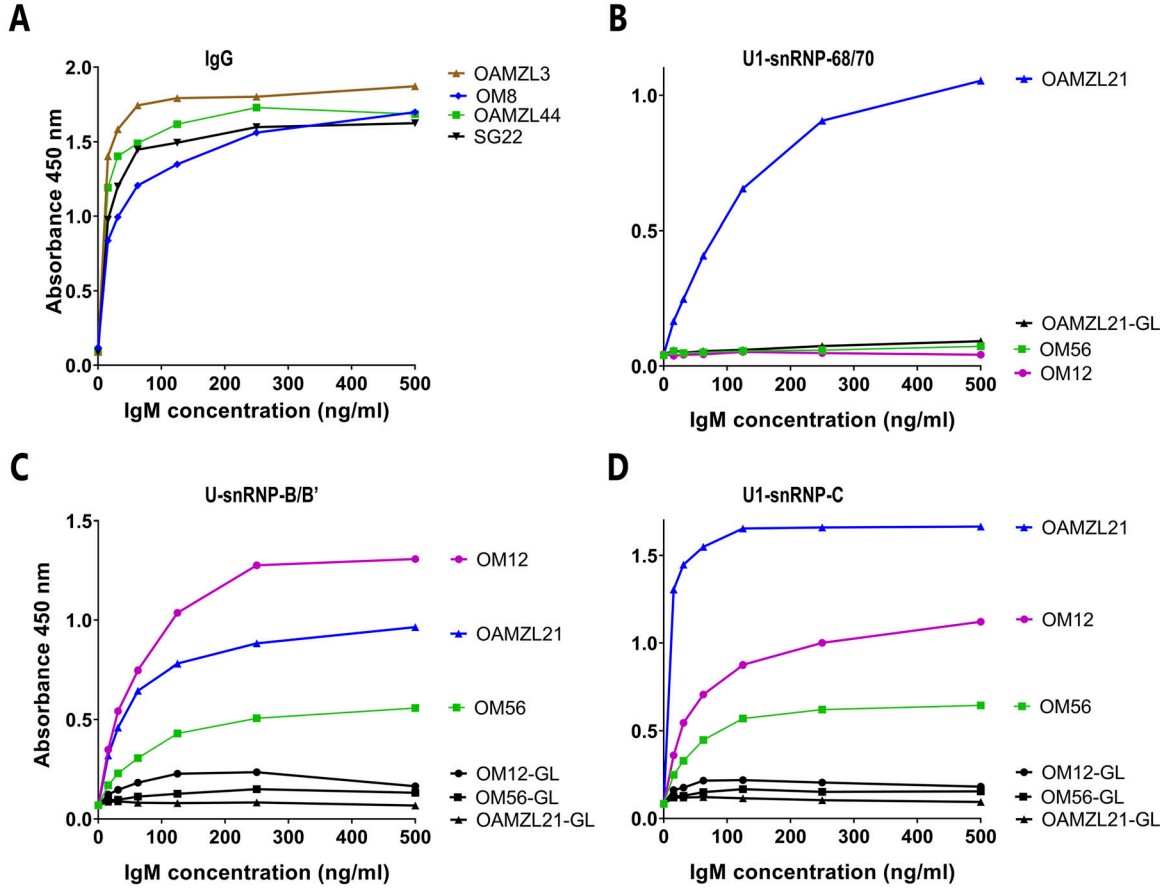

**Figure 4. Binding profiles of OAML IgMs to IgG and to U1-/U-snRNP complex proteins.**
**(A)** Specific binding of IgMs derived from OM8, OAMZL3, and OAMZL44 to human IgG (RF activity). SG22 is a control RF IgM derived of a Sjögren's syndrome patient. **(B, C, D)** Specific binding of IgMs derived from OM12, OM56, and OAMZL21 to U1-snRNP-68/70, U-snRNP-B/B', and U1-snRNP-C. OM12-GL, OM56-GL, and OAMZL21-GL denote IgMs in which the IGHV somatic mutations were reverted to their respective germline configuration.

the binding to the proteins of the U1/U-snRNP complexes. Interestingly, the OM12 IgM, which also reacted with NAL epitopes, lost binding to the U1-/U-snRNP proteins but displayed stronger binding to B cells after removal of its somatic IGHV mutations. This suggests that during development of the OM12 precursor B-cell clone, the process of affinity maturation favoring U1-/U-snRNP protein reactivity had resulted in a decreased binding with NAL epitopes. As OAML 38706 and OM12 express a highly homologous IGHV4-34-encoded BCR, it is very likely that OAML 38706 and OM12 also display a near-identical antigen binding profile specific for the U1-/U-snRNP complex. This and the finding that two stereotyped BCR CLL groups sharing VH-CDR3 homology with OAMZL12 and OAMZL26 both were also reactive with U1-/U-snRNP complex proteins strongly suggests that this particular form of autoreactivity may be more common and occur in a variety of B-cell malignancies.

Auto-antibodies specific for various proteins of the U1-/U-snRNP complexes, including U1-snRNP-A, C and 68/70 as well as the Sm proteins B/B', D1, D2, D3, E, F, and G, which form a ring around all U-snRNPs, are present in 20–40% of systemic lupus erythematosus patients (Dema & Charles, 2016). Moreover, the presence of antibodies specific for the U1-snRNP-associated proteins is a diagnostic criterion for mixed connective tissue disease (Dema &

Charles, 2016). Patients OM12 and OM56 of our hospital were not documented with an autoimmune disease. However, both patients suffer from type 2 diabetes (T2D), which in some studies has been associated with a higher prevalence of anti-nuclear antibodies. Moreover, a correlation between T2D and lymphoma has been suggested (Castillo et al, 2012; Litwińczuk-Hajduk et al, 2016).

It has been argued that RF-expressing MALT lymphomas experience chronic NF-κB pathway signaling as a result of dual RF-BCR/TLR stimulation, for example, in SS sialadenitis by IgG-SSA/SSB complexes containing single-stranded RNA stem-loop structures as TLR7 ligands (Bende et al, 2009, 2020, 2023). Likewise, in OAMLs, BCR/TLR7 co-stimulation may be mediated by U1-/U-snRNP complex proteins and associated U-RNA structures (Savarese et al, 2006).

## Materials and Methods

### Patient material

All OAMLs were diagnosed at the department of Pathology, Amsterdam University Medical Centers, The Netherlands. Here, we

**A**

| OAML/CLL | IGHV | VH-CDR3 region | Genbank/Reference |
|---|---|---|---|
| OAMZL12 | V2-5/D6-9/JH3 | C **AHR**QMYNSDWNGG**V**LDV WGQG | AFC97614 |
| CLL406 (subset #148B) | V2-5/D3-10/JH4 | C **AHR**REGNYGWDVAYLDY WGQG | A |
| CLL409 (subset #148B) | V2-5/D3-10/JH4 | C **AHR**LRMGAPWSWGTFD- WGQG | A |
| CLL410 (subset #148B) | V2-5/D3-3/JH4 | C **AR**RRERFSQWLSGDFDY WGQG | A |
| | | | |
| OAMZL26 | V3-66/D3-3/JH6 | C **ARG**HYDFWRA--TYYYYYGMDV WGQG | AFC97628 |
| CLL403 (subset #31) | V3-48/D3-3/JH6 | C **AR**DC-DFWSGYYGYYYYYGMDV WGQG | A |
| | | | |
| OAMZL37 | V4-30.4/D2-2/JH6 | C **ARD**PAANYYYGMDV WGQG | AFC97639 |
| CLL415 (subset #202) | V3-33/D4-17/JH3 | C **ARG**PSGDYVFAFDI WGQG | A |
| CLL416 (subset #202) | V3-30/D4-17/JH3 | C **A**QGAGGDYVFAFDI WGQG | A |
| | | | |
| OAML 50620163 | V3-7/JH4 | C **ARD**YYFDF- WGQG | AER35986 |
| CLL420 (subset #273) | V3-23/D4-17/JH5 | C **ARG**YFGDYN WGQG | A |

**B**

| Patient / CLL BCR subset | IGHV | IG(K/L)V | Conclusion Reactivity | Self/Auto antigens | | | | | | | | | | IA | | |
|---|---|---|---|---|---|---|---|---|---|---|---|---|---|---|---|---|
| | | | | Actin | Insulin | Human IgG (RF-activity) | SSA/Ro52 | SSA/Ro60 | U1-snRNP-68/70 | U1-snRNP-A | U-snRNP-B/B' | U1-snRNP-C | 5 other Auto Antigens | 6 fungal/Bacterial Antigens | 10 viral Antigens (Luminex) | 17 viral Antigens (Array) |
| CLL403 / #31 | V3-48/D3-3/JH6 | VL1-44/JL3 | **U-snRNP reactive** | green | green | green | green | green | red | red | red | red | green | green | green | green |
| CLL406 / #148B | V2-5/D3-10/JH4 | VL2-14/JL3 | **U-snRNP reactive** | green | green | green | green | green | green | orange | orange | orange | green | green | green | green |
| CLL409 / #148B | V2-5/D3-10/JH4 | VL2-14/JL3 | **U-snRNP reactive** | green | green | green | green | green | green | green | red | red | green | green | green | green |
| CLL410 / #148B | V2-5/D3-3/JH4 | VL2-8/JL3 | **U-snRNP reactive** | green | green | green | green | green | green | red | red | green | green | green | green | green |
| CLL415 / #202 | V3-33/D4-17/JH3 | VK3-15/JK1 | Non/Weak reactive | green | green | green | green | green | green | green | green | green | green | green | green | green |
| CLL416 / #202 | V3-30/D4-17/JH3 | VK3D-15/JK1 | Non/Weak reactive | green | green | green | green | green | green | green | green | green | green | green | green | green |
| CLL420 / #273 | V3-23/D4-17/JH5 | VK1-16/JK1 | Non/Weak reactive | green | green | green | green | green | green | green | green | green | green | green | green | green |
| | | | | | | | | | | | | | | | | |
| M8 | V3-30/D3-9/JH5 | VK1-39/JK4 | Non/Weak reactive | green | green | green | green | green | green | green | green | green | green | green | green | green |
| CLL57 (U) | V3-30.3/D3-3/JH6 | VL1-51/JL3 | **Strong Polyreactive** | red | red | red | red | red | red | red | red | red | red | red | red | red |
| CLL299 (U) | V1-2/D1-26/JH6 | VK4-1/JK2 | **Strong Polyreactive** | red | red | red | red | red | red | red | red | red | red | red | red | red |

**Figure 5. U1-/U-snRNP complex binding of stereotyped subset CLL IgMs with OAML VH-CDR3 homology.**
**(A)** VH-CDR3 homology between four OAMLs with BCR-stereotyped CLLs. Identical and similar amino acids are highlighted in red and blue, respectively. For OAMZL12, the identical phenylalanine (F) of CLL409 and CLL410 is highlighted in green. Reference A: PMID 32992344 (Agathangelidis et al, 2021). (B) Red and orange, respectively, indicates strong (>5 times background ABS 450 nm) and moderate (3–5 times background ABS 450 nm) antigen binding of the IgM in ELISA. Green indicates no antigen binding.

have selected 20 patients suffering from OAML, of which we have previously analyzed their clonal IGHV ("old" OAML cases) (van Maldegem et al, 2012) and 62 "new" OAML cases (OM32 – OM84, OM86 – OM94), which were collected between 2010–2021 at our department, originating from 32 males and 30 females (mean age 63 years, range 19–88 years). All cases consisted of CD20+ CD79A+ BCL2+ BCL6- small B cells, which were negative in split-fluorescence in situ hybridization for BCL2, BCL6 and MALT1 translocation, except for

one case OM86, which was MALT1 translocation positive. From the "old" study, we selected eight IgM/$\kappa$-expressing lymphomas (OM3, OM8, OM9B, OM12, OM23, OM24, OM30, and OM31), of which their IGKV chain was sequenced. From the "new" series we selected nine cases, of which frozen tissue was available and determined their IGHV and IGKV/IGLV sequences using RT–PCR (see below). Immunohistochemical analyses of these nine "new" cases showed that OM38, OM40, OM46, OM66, and OM68) expressed IgM/$\kappa$, OM67 and OM71 expressed IgG/$\kappa$, OM56 expressed IgA/$\lambda$ and of OM53 the staining results regarding the expressed IG isotype were inconclusive. IG clonality was confirmed by PCR using a fluorochrome (FAM)-labeled reverse primer for IGHJ, followed by PCR product length analysis (genescanning) (van Dongen et al, 2003). Primary chronic lymphocytic leukemia (CLL) samples, present in the series of the ERIC consortium (Agathangelidis et al, 2021) were obtained from the Erasmus Medical Center. In total, seven CLL samples were analyzed, that is, CLL403 (NL-01-0305-H1), CLL406 (NL-01-0005H1), CLL409 (NL-01-1260-H1), CLL410 (NL-01-0695-H1), CLL415 (NL-01-0530-H1), CLL416 (NL-01-1213-H1), and CLL420 (NL-01-1214-H1). CLL403 belongs to CLL BCR subset #31, CLL406, CLL409, and CLL410 belong to subset #148B, CLL415, and CLL416 belong to subset #202, and CLL420 belongs to subset #273. This study was conducted in accordance with the ethical standards of our institutional medical ethical committee on human experimentation and in agreement with the Helsinki declaration of 1975, as revised in 1983.

### Immunoglobulin sequencing and production of OAML-derived recombinant IgM antibodies

RNA was isolated from frozen sections using TRI Reagent (Merck; Sigma-Aldrich) and cDNA was synthesized using Pd(N)6 random primers and moloney murine virus RT (Thermo Fisher Scientific). The rearranged IGHV and IGKV/IGLV genes were amplified using IG family-specific leader primers in combination with a reverse primer, specific for the matching constant region of IgM, IgG, IgA, IgK, or IgL, respectively. The amplified IGHV, IGKV, and IGLV PCR products were sequenced using the Big Dye Terminator sequencing kit (Thermo Fisher Scientific). IG PCR products that yielded readable sequences were analyzed using the IMGT website with V-Quest (https://imgt.org/), as described before (Bende et al, 2020).

Based on the determined IGHV, IGKV, and IGLV sequences of the PCR products, appropriate IGHV/IGKV/IGLV family-specific leader primers were combined with an IGHJ/IGKJ/IGLJ primer to produce full length IG products including restriction sites, which were cloned in pTOPO-TA plasmids. Individual pTOPO plasmids were sequenced and compared with the IG sequences of the initial PCR products. pTOPO plasmids with consensus IG sequences were selected and these IG sequences were subsequently cloned in the pIGH($\mu$) and pIGL($\kappa$) expression plasmids, which were co-transfected in SP2/0 myeloma cells, as we have previously described (Bende et al, 2005). Supernatants of the transfected SP2/0 cells were screened for IgM/$\kappa$ production, using ELISAs.

IGHV and IGKV full-length sequences of OAMLs as determined by Zhu et al (2011, 2013) of OAMZL1, 3, 4, 9, 10, 13, 16, 21, and 44 and IGHV of OM12, OM56, and OAMZL21, in which somatic mutations were reverted to germline configuration were ordered at GenScript. Recombinant IgMs were produced as described above. Table S1

provides a complete overview of the patient material, the number of IGHV and IGKV/IGLV sequences and the number of recombinant IgMs produced.

### Binding experiments of OAML-derived IgM to erythrocytes and IgG-expressing B-cell lines

The EBV transformed memory B-cell line LOS2 (Bende et al, 1992) and the DLBCL cell line LY3, both IgG expressing, as well as Rhesus(D) positive and negative erythrocytes were incubated at a number of about $5 \times 10^5$ cells in 200 $\mu$l with recombinant OAML-derived IgM at 1 $\mu$g/ml for 30 min on ice. After washing the cells were stained with anti-human IgM-PE, 1:1,000 diluted (SouthernBiotech) for 30 min on ice, washed again and measured with a FACS apparatus (BD Biosciences).

### Screening of ocular adnexal MALT lymphoma-derived IgM for antigen binding

Antigen binding ELISAs were performed as described previously (Bende et al, 2005, 2016, 2020). Binding experiments of the OAML IgM, with viral antigens was performed in two laboratories using a Luminex assay (Grobben & Tejjani, Amsterdam UMC, department of Medical Microbiology and Infection Prevention) and an antigen array (Sikkema & Chandler, Erasmus MC, Rotterdam, department of Viro science) (Koopmans et al, 2012; Grobben et al, 2021; Claireaux et al, 2022; de Bellegarde de Saint Lary et al, 2023). The antigen preparations that were used are listed in Table S7. Immunohistochemistry on tissue microarrays (TMA) containing most normal human tissues were performed as described (Hoogeboom et al, 2013a).

# Supplementary Information

# Acknowledgements

We thank Kim Brandwijk from the department of Medical Biology of Amsterdam UMC for FACS assistance. Wim van Est from the department of Pathology of Amsterdam UMC for help in preparing the figures. This research was supported by grants of the Dutch Arthritis Foundation (2015-2-310) and of the Dutch Cancer Society (UVA2009-4524 and UVA2014-6824).

### Author Contributions

RJ Bende: conceptualization, resources, data curation, formal analysis, supervision, investigation, visualization, methodology, and writing—original draft, review, and editing.
N Donner: conceptualization, investigation, methodology, and writing—original draft, review, and editing.
TAM Wormhoudt: formal analysis, investigation, and methodology.
A Beentjes: formal analysis, investigation, and methodology.
A Scantlebery: formal analysis, investigation, and methodology.

M Grobben: data curation, investigation, and methodology.
K Tejjani: data curation, investigation, and methodology.
F Chandler: data curation, investigation, and methodology.
RS Sikkema: data curation, investigation, and methodology.
AW Langerak: resources, validation, and writing—review and editing.
JEJ Guikema: resources, validation, and writing—review and editing.
CJM van Noesel: conceptualization, resources, methodology, and writing—original draft, review, and editing.

## Conflict of Interest Statement

The authors declare that they have no conflict of interest.

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
