## [Reviewer comments · Life Science Alliance]

Life Science Alliance

Distinct groups of autoantigens as drivers of ocular adnexal MALT lymphoma pathogenesis

Richard Bende, Naomi Donner, Thera Wormhoudt, Anna Beentjes, Angelique Scantlebery, Marloes Grobber, Khadija Tejjani, Felicity Chandler, Reina Sikkema, Anton Langerak, Jeroen Guikema, and Carel van Noesel

DOI: <https://doi.org/10.26508/lsa.202402841>

Corresponding author(s): Richard Bende, Amsterdam University Medical Centers

Review Timeline:	Submission Date:	2024-05-23
	Editorial Decision:	2024-06-05
	Revision Received:	2024-06-17
	Editorial Decision:	2024-06-20
	Revision Received:	2024-06-28
	Accepted:	2024-06-28

Transaction Report:

Please note that the manuscript was previously reviewed at another journal and the reports were taken into account in the decision-making process at *Life Science Alliance*. Since the original reviews are not subject to Life Science Alliance's transparent review process policy, the reports and author response cannot be published.

June 5, 2024

Re: Life Science Alliance manuscript #LSA-2024-02841-T

Dr. Richard J Bende
Amsterdam University Medical Centers, Location AMC
Pathology
Meibergdreef 9
Amsterdam, North Holland 1105 AZ
Netherlands [NL]

Dear Dr. Bende,

Thank you for submitting your manuscript entitled "Distinct groups of autoantigens as drivers of ocular adnexal MALT lymphoma pathogenesis" to Life Science Alliance. We invite you to submit a revised manuscript addressing the Reviewer comments.

Thank you for this interesting contribution to Life Science Alliance. We are looking forward to receiving your revised manuscript.

Sincerely,

B. MANUSCRIPT ORGANIZATION AND FORMATTING:

June 20, 2024

RE: Life Science Alliance Manuscript #LSA-2024-02841-TR

Dr. Richard J Bende
Amsterdam University Medical Centers
Pathology
Meibergdreef 9
Amsterdam, North Holland 1105 AZ
Netherlands

Dear Dr. Bende,

Thank you for submitting your revised manuscript entitled "Distinct groups of autoantigens as drivers of ocular adnexal MALT lymphoma pathogenesis". We would be happy to publish your paper in Life Science Alliance pending final revisions necessary to meet our formatting guidelines.

- please be sure that the authorship listing and order is correct
- please add the Twitter handle of your host institute/organization as well as your own or/and one of the authors in our system
- please remove information from the title page, such as Revised version 1, number of words...etc
- please consult our manuscript preparation guidelines <https://www.life-science-alliance.org/manuscript-prep> and make sure your manuscript sections are in the correct order
- please use the [10 author names et al.] format in your references (i.e., limit the author names to the first 10)
- please add a Conflict of Interest statement to your main manuscript text
- please add your main, supplementary figure, and table legends to the main manuscript text after the references section
- please remove figures from the manuscript file. They should stay uploaded separately
- for publication of figures, we require PowerPoint, TIFF, PDF, or EPS files, so please upload supplementary figures in one of these formats
- please add callouts for Figure 2A-B and Table S8 to your main manuscript text

A. FINAL FILES:

B. MANUSCRIPT ORGANIZATION AND FORMATTING:

Sincerely,

June 28, 2024

RE: Life Science Alliance Manuscript #LSA-2024-02841-TRR

Dr. Richard J Bende
Amsterdam University Medical Centers
Pathology
Meibergdreef 9
Amsterdam, North Holland 1105 AZ
Netherlands

Dear Dr. Bende,

Thank you for submitting your Research Article entitled "Distinct groups of autoantigens as drivers of ocular adnexal MALT lymphoma pathogenesis". It is a pleasure to let you know that your manuscript is now accepted for publication in Life Science Alliance. Congratulations on this interesting work.

DISTRIBUTION OF MATERIALS:

Again, congratulations on a very nice paper. I hope you found the review process to be constructive and are pleased with how the manuscript was handled editorially. We look forward to future exciting submissions from your lab.

Sincerely,
